# Arrhythmic Mitral Valve Prolapse: A Comprehensive Review

**DOI:** 10.3390/diagnostics13182868

**Published:** 2023-09-06

**Authors:** Yuyan Deng, Jinfeng Liu, Shan Wu, Xiaoming Li, Huimei Yu, Lili Tang, Meng Xie, Chun Zhang

**Affiliations:** Department of Interventional Ultrasound, Beijing Anzhen Hospital, Capital Medical University, Beijing 100029, China; dengyuyan1019@163.com (Y.D.); liujinfeng1990@163.com (J.L.); mountwu2016@sina.com (S.W.); spirit1701@sina.com (X.L.); xie.hong1979@163.com (H.Y.); tanglili0112@sina.com (L.T.); xiemengcelia@163.com (M.X.)

**Keywords:** arrhythmic mitral valve prolapse, mitral annular disjunction, ventricular arrhythmias, echocardiography, cardiac magnetic resonance

## Abstract

Mitral valve prolapse (MVP) is a prevalent cardiac disorder that impacts approximately 2% to 3% of the overall population. While most patients experience a benign clinical course, there is evidence suggesting that a subgroup of MVP patients face an increased risk of sudden cardiac death (SCD). Although a conclusive causal link between MVP and SCD remains to be firmly established, various factors have been associated with arrhythmic mitral valve prolapse (AMVP). This study aims to provide a comprehensive review encompassing the historical background, epidemiology, pathology, clinical manifestations, electrocardiogram (ECG) findings, and treatment of AMVP patients. A key focus is on utilizing multimodal imaging techniques to accurately diagnose AMVP and to highlight the role of mitral annular disjunction (MAD) in AMVP.

## 1. Introduction

Mitral valve prolapse (MVP) is a prevalent cardiac disorder that impacts approximately 2% to 3% of the overall population, with a similar prevalence in males and females [1,2,3]. It is typically characterized by a superior displacement of the mitral leaflet(s) into the left atrium (LA) during ventricular systole [4,5,6]. While numerous studies have considered the vast majority of MVP to be benign [7,8], the prognosis for MVP varies widely and is an ongoing debate [9,10].

MVP can lead to serious complications such as significant mitral regurgitation (MR), atrial fibrillation, bacterial endocarditis, congestive heart failure, stroke, ventricular arrhythmia (VA), and even sudden cardiac death (SCD) [1,2,11,12,13].

As a result, the term “arrhythmic mitral valve prolapse (AMVP)” has been introduced, sparking a debate about its definition and clinical significance in terms of diagnosis, risk stratification, and treatment [14,15]. At the same time, concerns have been raised about the link between MVP and SCD [15,16,17,18]. The incidence of SCD due to MVP in the general population is typically very low, and due to the lack of uniform diagnostic criteria, MVP is not even categorized as a contributing factor to SCD [16]. However, a meta-analysis by Han demonstrated that the overall prevalence of SCD among MVP cases was 217 incidents per 100,000 person years, which is significantly higher when contrasted with 42–53 incidents per 100,000 person years observed within the broader populace [17].

This review provides a comprehensive overview of the historical background, epidemiology, pathology, clinical manifestations, characteristics of electrocardiogram (ECG), and mechanism of VA occurrence concerning AMVP. Moreover, we delve into the imaging diagnosis of AMVP, highlighting the distinctive echocardiography, computed tomography (CT), and cardiac magnetic resonance (CMR) characteristics while also summarizing the associated risks of each imaging technique. By carefully assessing these risks, we aim to present diagnostic tools that could be used in risk stratification for the better management and prevention of MVP-related SCD.

### 1.1. Historical Background

Barlow first discovered the mitral valve origin of the late systolic murmur commonly associated with clicks by using a left ventricular cinematic angiography in the 1960s [19]. Then, Criley termed this symptom MVP [20]. The fact that MVP can cause VAs has been noted since the 1970s [21,22], but Basso formally introduced the concept of AMVP in 2015 [22]. More recently, the *EHRA expert consensus statement on arrhythmic mitral valve prolapse and mitral annular disjunction complex* states that the diagnostic requirements for AMVP are the presence of MVP (with or without mitral annular disjunction (MAD)), combined with frequent (≥5% total premature ventricular contraction (PVC) burden) and/or complex VA (ventricular tachycardia (VT), non-sustained ventricular tachycardia (NSVT) and ventricular fibrillation (VF)), in the absence of any other definitive arrhythmic substrate [23].

### 1.2. Epidemiology

After reviewing some case reports, autopsy studies, and clinical studies, it has been suggested that AMVP has a higher prevalence in females [14,17,24]. However, this observation could be attributed to a publication bias, survivorship bias, and referral bias. For instance, a nationwide US autopsy study conducted between 2000 to 2018 found that gender distribution was equal [25]. Based on a larger cohort study, it seems that AMVP can affect individuals of varying ages and genders [26].

As of now, the prevalence of AMVP has not been fully defined. Several reasons contribute to the differences in the prevalence of VAs in MVP, including the use of multimodal imaging, which results in varying definitions of mitral valve patterns, the heterogeneity of the study population, the complexity of VAs, and the absence of a systematic assessment of long-term ECG records [12,14,15,26].

### 1.3. Pathology

MVP can manifest either as an isolated mitral valve abnormality or as part of a well-defined syndrome of heritable connective tissue disorders, such as the Marfan syndrome, polycystic kidney disease, Ehlers–Danlos syndrome, etc. [27,28]. Pathologically, MVP can be categorized into Barlow’s disease (characterized by myxoid infiltration and thickening of the mitral valve leaflet, often leading to multi-segmental prolapse) or fibroelastic deficiency (damage to connective tissue, resulting in leaflet and chordae thinning, and even rupture) [29].

Efforts have been made to study the pathological changes associated with AMVP. Some studies have revealed that AMVP is typically characterized by severe myxomatous degeneration, accompanied by significant leaflet redundancy, as well as excess leaflet length and thickness. These morphological changes are considered to be the hallmarks of AMVP [15,28]. Sriram et al. published a noteworthy study on 10 cases of MVP patients who survived cardiac arrest (CA) despite the absence of identifiable electrical or structural heart disease [14]. Importantly, all of these patients exhibited a distinctive characteristic known as bileaflet mitral valve prolapse (BiMVP). Consequently, the researchers introduced the term “Malignant Bileaflet Mitral Valve Prolapse Syndrome” to specifically identify and differentiate this subgroup of MVP patients [14]. However, the results of Nordhue’s retrospective study suggested that BiMVP, while associated with VT, does not seem to forecast an unfavorable prognosis on a broader population scale unless accompanied by other risk factors. The cohort study included 18,786 patients who underwent echocardiography at the Mayo Clinic for 14 years and were divided into the BiMVP group, the single-leaflet mitral valve prolapse (SiMVP) group, and controls without MVP. Statistical analysis showed that the BiMVP group had a high incidence of VAs. Nevertheless, there were no noteworthy disparities in the overall mortality and implantable cardioverter defibrillator (ICD) implantation rate among all groups [30].

In anatomical studies, the normal mitral annulus is always described as a ‘D-shaped’ structure, with the anterior and posterior mitral valve leaflets inserted [31,32]. However, scientists have noticed an abnormal attachment of the posterior leaflet, which is directly on the atrial wall, called MAD [33,34]. Through further studies, MAD has been described as an anomalous detachment of the mitral annulus from the myocardium of the basal left ventricular free wall, accompanied by an atypical systolic excursion of the leaflet hinge point of the posterior reaching into the LA [33,34,35,36] (Figure 1). The mitro-aortic fibrous continuity linking the anterior mitral leaflet and the aortic cusps restricts the circumferential extension of MAD. Therefore, MAD is only observed at the insertion of the posterior leaflet, and this abnormal disjunction extends laterally in a variable manner beneath all scallops of the posterior leaflet, with a notable emphasis on the central posterior scallop [37,38].

Numerous investigations have demonstrated a substantial link between MAD and AMVP [26,36,39,40]. These results confirmed that AMVP patients exhibited a greater incidence of MAD compared to non-arrhythmia MVP patients [26,41]. Interestingly, even without MVP, MAD seems to be associated with complex arrhythmia events (aborted cardiac arrest (CA) and VT), suggesting that MAD itself may be a marker of complex VAs [35].

Abnormal posterior mitral annulus systolic curling was initially described among MVP patients during the 1970s [42] and is marked by an abnormal or excessive systolic upward movement of the posterior mitral annulus along with the contiguous posterobasal myocardium. This phenomenon involves the significant thickening and hypertrophy of the basal myocardial wall, which has also recently been demonstrated to have a robust connection with MAD [36]. The imaging features of MAD are carefully described in this paper later.

### 1.4. Clinical Manifestations

MVP can present with a variety of non-specific symptoms, including atypical chest pain, exertional dyspnoea, palpitations, and syncope [11,12]. The first three symptoms are common but with comparable proportions found in MVP patients, regardless of whether they have experienced VAs [43,44].

Nevertheless, syncope is uncommon in MVP patients without VAs [1,43] and was documented in 35% of patients with MVP and SCD or CA [17]. In addition, syncope was more likely to occur in patients with more severe VAs in a substantial gathering of consecutive MVP patients [26]. Hence, syncope, particularly unexplained syncope, holds a significant discriminative value in recognizing MVPs who are susceptible to VAs.

### 1.5. Electrocardiography

Electrocardiography serves as a valuable and cost-effective initial diagnostic instrument for detecting AMVP patients. The majority of MVP patients exhibit unremarkable findings in their resting ECG results. Nonetheless, individuals who have experienced CA or SCD frequently present with abnormalities in both their resting ECG and their 24 h Holter ECG [15,26].

Repolarization abnormalities are generally considered a marker of arrhythmia risk in MVP patients. In MVP-related SCD patients, a characteristic abnormal T wave on the ECG may be observed, such as a T-wave inversion (TWI) or biphasic T waves, typically in the inferior and lateral leads [15,17,26]. Studies have found that VAs can also be caused by prolonged QT intervals, and QT prolongation is correlated with the thickening of the anterior leaflet and a more severe leaflet prolapse [45]. However, relevant research evidence is currently limited, and the correlation between the prolonged QT interval and MVP mortality remains uncertain [46,47]. Fragmented QRS has been identified as a marker of a localized scar linked to complex VAs in individuals diagnosed with MVP [48].

PVC and VT are also common in MVP patients, who tend to exhibit pleomorphic PVCs more frequently than the general population, and those with MR tend to show more PVCs than those without [26,49]. Surprisingly, most cases of SCD in relation to MVP occur in individuals with only mild or moderate MR as opposed to severe MR. The severity of MR does not appear to be linked to the arrhythmias burden [50]. MVP-related VT appears to manifest with greater frequency in females, although the precise cause for this divergence remains unclear and may be multifactorial [51].

### 1.6. Mechanism of VAs Occurrence

In 1980, Lichstein [52] conducted a study on the vectorcardiogram of PVCs in MVP and discovered that the most frequent site of origin for these arrhythmias was the posterior basal portion of LV. This observation supports the hypothesis that the mechanical irritation of the valve, which is responsible for pumping blood into the LV, could potentially trigger VAs.

More recently, Sriram et al. noted a PVC configuration outflow tract origin alternating with the papillary muscle (PM) or fascicular origin [14]. Outflow tract PVCs always originate from LV and, in some instances, from both the left ventricular and right ventricular outflow tracts. Electrophysiologic studies (EPS) have also identified the PMs, the left ventricular outflow tract, and the mitral annulus as the original sites of these arrhythmias, suggesting that PVCs emerging in proximity to the prolapsing leaflet and its neighboring structures are the underlying cause of arrhythmias [14].

Scientists have attributed SCD to VF [53,54]. However, the specific relationship between MVP and VF has not been fully elucidated, and currently, experts believe that anatomical substrates and triggers play a role in SCD occurrence.

The histological evidence of the myocardial substrate in MVP-related SCD patients has revealed the presence of patchy fibrosis between the mitral valve, PM, and the adjacent inferobasal left ventricular myocardium [15,55]. This fibrosis may result from the mechanical pulling of the prolapsed mitral valve leaflet [56]. Continuous mechanical pulling of prolapsed leaflets has the potential to induce myocardial hypertrophy and fibrosis in the basal and inferolateral ventricles, as well as the PMs that support the mitral valve structure [57,58]. Second, friction between chordae and the left ventricular mural endocardium may cause endocardial friction lesions, which, in some cases, can lead to significant fibrosis in the mural endocardium. The mechanical stimulation of the endocardium using thickened chordae might be responsible for VAs and repolarization changes [59]. Mechanically induced left ventricular fibrosis serves as a substrate for VAs [60], where z slow and heterogeneous conduction associated with fibrosis may facilitate the instauration of re-entrant circuits, thereby increasing their vulnerability to VT/VF [61,62].

Most MVP patients who have experienced SCD have a history of PVCs. Typical PVCs in MVP patients frequently originate from the PM, fascicular system, and outflow tract. These PVCs result from the acute mechanical stretching of damaged tissues, which shortens the potential duration of the action and decreases diastolic potential. This process leads to stretch-activated early afterdepolarization, ultimately triggering PVC activity [63,64]. However, monomorphic or polymorphic PVCs are not enough to indicate elevated SCD risk in MVP patients; however, the ectopic origin point of PVCs may be an important marker of the occurrence of SCD. In fact, distal Purkinje fibers are prone to depolarization and automation anomalies, and in Syed FF‘s experiments, for the six patients with BiMVP who underwent catheter ablation, Purkinje-originated PVCs were all found to be VF induction sites and had CA disease history [65]. Therefore, the Purkinje system plays a key component in VA onset and SCD risk [65,66].

## 2. Multimodality Imaging of MVP

### 2.1. Echocardiography

Echocardiography can be used to diagnose MVP, follow up, and evaluate whether to intervene [11], which is the foremost and frequently employed imaging modality for MVP (Figure 1). MVP represents an abnormal systolic change in the mitral valve as a displacement of one or both of its leaflets at more than 2 mm concerning the plane passing through the mitral valve annulus, superiorly into the LA, which can be evaluated in the parasternal long-axis or the apical 3-chamber view on 2D transthoracic echocardiography (TTE) [67,68,69].

We need to know that the apical four-chamber view cannot be employed in diagnosis. Because the mitral valve is a saddle-shaped structure, its lowest points (closest to the apex) are located medially and laterally, while its highest points are located anteriorly and posteriorly at the aortic insertion and left ventricular posterior wall [70]. Thus, when leaflet(s) appear above the mitral annulus on the apical four-chamber section, there may not be a distortion in the leaflet or genuine displacement over the entire mitral valve [70], potentially resulting in a misdiagnosis.

Experienced physicians can effectively utilize 2D TTE to accurately identify the specific scallop(s) of lesions in most MVP patients. In the parasternal long-axis view, various measurements can be obtained, including the mitral annulus diameter, anterior and posterior leaflets length, and thickness, as well as the leaflets’ closing position relative to the mitral annulus line. Furthermore, TTE also serves as the primary method for the thorough assessment of MR associated with MVP and allows for the precise quantification of MR through a variety of methods, such as the vena contracta, the proximal isovelocity surface area, and the MR regurgitant volume and fraction measured by continuous-wave Doppler and classified as trace-mild (regurgitant volume ≤ 15 mL), moderate (16–59 mL), or severe (≥60 mL) [71,72]. The duration of MR is a crucial factor to consider. In MVP patients, the jet area and effective regurgitant orifice assessed during mid-late systolic MR can be similar to those observed in holosystolic MR. However, the mid-late systolic MR results in a lower regurgitant volume and more benign outcomes [73]. At the same time, TTE quantifies the dimensions and function of LA and LV, both of which are affected by clinically significant MR [7].

Compared with 2D echocardiography, 3D echocardiography facilitates the differentiation between leaflet prolapse and billowing (Figure 2). It provides a more accurate understanding of the mitral valve’s structure and function, the specific location of the prolapse, and whether the commissure is involved, and can accurately define and measure the volume of the ventricle and the understanding of mitral annulus dynamics and whether MR pathophysiology has improved, especially using real-time 3D transesophageal echocardiography (TEE) [7,74,75]. The advent of 3D technology has also led to significant improvements in the accuracy of the MR severity assessment. Finally, 3D TEE is considered critical for the pre- and intra-procedural evaluation of mitral valve anatomy and MR mechanisms and is widely used [76].

With the continuous advancements in echocardiographic techniques, several abnormalities related to VAs can now be identified, including mitral annular tissue Doppler velocity, mechanical dispersion, a global longitudinal strain, and post-systolic shortening. These novel imaging modalities have shown promise in small-scale studies, providing valuable insights into AMVP. However, it is important to note that further validation and larger-scale investigations are imperative prior to their wide adoption in clinical practice.

Valvular prolapse, MAD, and systolic curling are closely associated with the excessive contraction of the anterolateral, posterior, and inferior wall basal segments of the LV. The prolapsing leaflets stretch the myocardium, triggering a sudden increase in tissue velocity that abruptly impacts the PMs and adjacent myocardium. Enhanced tissue velocities can be gauged using tissue Doppler imaging through a standard apical window. The probe is aligned based on the motion of the mitral annulus; when the peak velocity is ≥16 cm/s, it is termed the “Pickelhaube sign”. Recent studies have revealed that patients exhibiting the “Pickelhaube sign” have a higher incidence of malignant VAs, which may be a risk marker for AMVP [77,78,79].

Speckle tracking echocardiography stands as an advanced echocardiographic technique that provides a detailed analysis of the myocardial strain across diverse myocardial segments, which can accurately measure myocardial segmental systolic deformation. Several studies have elucidated an association between MVP and heterogeneity in the left ventricular strain, notably with higher strains observed in the left ventricular basal and posterolateral segments. However, its potential arrhythmia-inducing role remains controversial [58,80]. The detection of parameter changes in speckle-tracking echocardiography is particularly valuable because they may precede more conventional cardiac function changes [81] and help assess the degree of myocardial stretch in instances of significant primary MR.

Left ventricular mechanical dispersion represents another innovative parameter in speckle-tracking analysis; characterized as the standard deviation of the peak longitudinal strain time in all left ventricular segments, it has the potential to aid in the identification of MVP patients who are at an increased risk of experiencing VAs [82]. Mechanical dispersion was observed to exhibit higher values among AMVP patients than those without arrhythmias [83]: a marker of heterogeneous ventricular contractions [82].

The post-systolic index is a parameter calculated by the following formula: [(maximum strain % in cardiac cycle-peak systolic strain %)/(maximum strain % in cardiac cycle)] × 100. It quantifies the degree of myocardial shortening that occurs after the peak systolic strain during the cardiac cycle. It has been reported that post-systolic shortening in two or more myocardial segments could be associated with increased mortality and morbidity [84]. In a recent study, AMVP patients were found to have higher regional post-systolic shortening in the basal to mid-lateral segments compared to those with N-AMVP [85]. This finding suggests that post-systolic shortening in specific myocardial regions may serve as an indicator of adverse outcomes in MVP patients.

TTE and TEE are routine tools in practice [71,72]. Nonetheless, echocardiography does possess some inherent constraints as it relies on the operator, the size of the patient, and instrumental settings [86].

### 2.2. Cardiac Magnetic Resonance

CMR is a non-invasive imaging technique with unlimited imaging planes, which can provide reproducible quantitative data on cardiac structure and function [87,88].

CMR allows for precise measurements of the left ventricular mass and end-diastolic thickness. This encompasses gauging the ratio of the basal-to-mid left ventricular wall thickness, identifying MAD and systolic curling (presence or absence) and quantifying it, measuring the mitral valve leaflet length, leaflet diastolic thickness, annulus diameter (at end-diastole and end-systole in both inter-commissural and anteroposterior aspects), and prolapse distance [89,90] (Figure 3).

More importantly, CMR can help determine the myocardial composition and identify VAs’ specific risk factors, assess myocardial fibrosis/scar, describe its location, and quantify it [91]. Late gadolinium enhancement (LGE) [92,93] sequences reveal patchy macroscopic fibrosis, while T1 mapping identifies diffuse fibrosis at a microscopic level (Figure 3).

Han et al. [89] first exhibited in 2008 that CMR can identify MVP using identical echocardiographic criteria and found that MVP patients had increased blood-to-leaflet contrast ratios, longer and thicker posterior leaflets, and enlarged MADs. Also, it can ideally identify specific arrhythmogenic fibrotic regions in MVP patients.

Following their identification of fibrotic changes through post-mortem examinations, Basso et al. hypothesized that these alterations could be discerned in vivo through CMR. They performed CMR and histopathological examinations on a group of 43 instances of SCD involving young patients with MVP. The results indicated the presence of PM fibrosis (88%) or inferobasal fibrosis (93%). LGE distribution on CMR exhibited heightened gadolinium in pathological tissue relative to healthy tissue, which is associated with histopathological fibrosis [15].

This insight was validated in another 3680 autopsy patients, 62 of whom were MVP (1.7%), 74% of whom had myocardial fibrosis involving one or both PMs (mainly posteromedial PM) and the adjacent left ventricular wall (mainly posteroinferior wall of LV) [94].

Another study found that compared with non-MVP-related MR, CMR in MVP patients showed significant left ventricular fibrosis, and the left ventricular fibrosis increased with the severity of MR, which tended to localize in specific regions of the LV and was associated with sustained VT or VF. Among them, the incidence of left ventricular fibrosis was 36.7% in MVP patients and only 6.7% in non-MVP patients. In addition, fibrotic myocardium was most commonly found in the inferior basal wall in MVP patients (31.1%). This suggests that there is a unique pathophysiology in MVP patients beyond volume overload. Left ventricular fibrosis, in the case of primary MR, could potentially serve as an indicator of heightened risks for arrhythmic events [95].

The EHRA expert consensus has concluded that CMR should be performed in all individuals who have experienced sustained VAs or survived SCD before the implantation of an ICD [23]. This is because CMR can enhance the rate of etiological diagnosis in MVP patients with a history of survived SCD or sustained VT while excluding other etiology [96]. We further recommend conducting a CMR examination on individuals with a background of documented NSVT or an unexplained syncope [26]. In addition, it may be valuable to perform a CMR on patients whose echocardiographic images are not clear enough to accurately assess structural changes, mitral valve characteristics, or left and right ventricular function [23].

### 2.3. Computed Tomography

To date, only a limited number of studies have evaluated the diagnostic accuracy of CT for MVP [86,97]. CT is primarily employed for diagnosing coronary artery disease [98,99], but it has been observed in several studies that CT is effective when evaluating valve morphology and function [100,101]. Corresponding with echocardiography findings, the combination of two-chamber and three-chamber views is the most accurate diagnosis for MVP with CT, while the use of the four-chamber view led to the overestimation of billowing, possibly because the normal mitral valve is saddle-shaped [102]. Because of the high sensitivity of cardiac CT, it can also measure leaflet thickening, which is deemed an additional criterion for diagnosing classic MVP (Barlow’s disease) [103]. Cardiac CT exhibits a sensitivity of approximately 84.6% and a specificity of 100% when evaluating abnormalities relating to the mitral valve [103].

Although cardiac CT demonstrates outstanding diagnostic accuracy in identifying the presence of MVP in patients, the limitations of cardiac CT for mitral MVP assessment include concerns about radiation exposure, a suboptimal temporal resolution, and a constrained capability to thoroughly examine each leaflet [104].

### 2.4. Mitral Annular Disjunction

MAD occurrence is associated with the presence of advanced myxomatous disease, which is characterized by significant leaflet redundancy and BiMVP. MAD causes annulus and left ventricular de-anchoring [39], leading to an inefficient ventricular contraction and an increase in left ventricular dimensions during the end-systole. These changes may carry prognostic implications for individuals with AMVP [105]. The measurement of the MAD trench is typically performed at the end-systole via the parasternal or apical long-axis view through TTE. Cardiac CT and CMR are capable of diagnosing MAD as well, acting as complementary diagnostic instruments for patients with a minor degree of MAD [106,107]. Additionally, it is worth noting that a wider MAD trench has been linked to a heightened incidence of NSVT detected during 24 h Holter ECG monitoring [34].

According to Maurice Enriquez-Sarano [108], MAD cannot be diagnosed during diastole because, at that phase, the posterior annulus (i.e., the hinge line of the posterior leaflet) appears to be in a normal position, sitting on the ventricular myocardium as the myocardium relaxes. However, during systole, the posterior myocardium contracts, causing the hinge line to “slide” backward, causing the MAD to be visible and separate from the ventricular myocardium [31,37]. However, there is still uncertainty about the specific structure or tissue that fills the gap between the crest of the ventricular myocardium and the hinge line [10]. In some cases, the abundance of the posterior leaflet tissue, stemming from a normally positioned annulus, might potentially be misdiagnosed as MAD without dynamic examination. Therefore, both echocardiography and CMR require dynamic analysis to avoid misinterpretation and ensure the accurate characterization of MAD and MVP [23].

At present, the definition of MAD has not yet determined the threshold, and the threshold is set arbitrarily in most studies [35,38,41]. For example, Dejgaard et al. defined MAD as ≥1.0 mm in CMR [35]. However, Toh et al. found that the median maximum separation height on normal CT was 3.0 mm [107]. Obviously, choosing a higher threshold results in a lower calculated prevalence and degree of MAD. Apart from discrepancies in defining the disjunction (with or without specified threshold values), several other factors could contribute to the diverse results observed in different studies. These factors include differences in the patient cohorts studied, the spatial resolution of measurements acquired, the assessment approach (whether 2D or 3D), and the type of samples used (cadaveric or living heart) [107].

### 2.5. Risk Stratification

Risk stratification in MVP is a complex task due to the absence of a single risk factor that reliably predicts malignant VAs and SCD. Furthermore, the combination of risk factors that elevate the risk in MVP patients remains unclear. This presents the challenge of identifying high-risk individuals within a larger population of low-risk patients (Figure 4). Multiple studies have demonstrated that certain factors, such as myxomatous degeneration accompanied by significant leaflet redundancy, as well as excess leaflet length and thickness, experiencing complex VAs, demonstrating MAD, and exhibiting LGE on CMR, could help distinguish between low-risk and high-risk MVP patients [9]. It is noteworthy that while there is an established link between MVP and SCD, the latest ESC Guidelines do not provide recommendations for the risk stratification of SCD in MVP patients [109].

The EHRA expert consensus classifies MVP-related VAs into low, intermediate, and high-risk groups, which are aligned with a broader classification of VAs (Table 1). As VAs can develop secondarily, it is necessary to establish a plan for re-evaluation over time, with the intensity and frequency of such assessments depending on the existence and quantity of phenotype risk features. For instance, patients experiencing an unexplained syncope are well-suited candidates for extended ECG monitoring through longer periods or implantable loop recorders, even if no VT is evident on Holter monitoring. Conversely, asymptomatic MVP patients lacking complex VAs in their initial screening may require only an episodic Holter assessment, with increased frequency if they exhibit phenotype risk features. Currently, there is no definite assessment frequency for the reclassification of VA risk, but periodic Holter monitoring might help achieve this objective [23].

### 2.6. Management

According to the EHRA expert consensus [23], there are currently four common options considered for preventing SCD in MVP patients. These options include medical therapy, catheter ablation, ICD implantation, and mitral valve surgery. The treatment options for AMVP are aimed at alleviating symptoms and improving survival.

AMVP patients are typically prescribed the same medications for VAs as other patients, such as calcium channel blockers, β blockers, and other antiarrhythmic drugs [110]. However, there is limited evidence to support their use in improving survival, specifically for AMVP.

Catheter ablation is an effective method for treating scar-related malignant arrhythmias [65,111]. PVC triggers, frequently situated on PMs, are capable of being accurately mapped and successfully ablated, improving symptoms and reducing ventricular ectopic burden [65].

The current evidence is insufficient to support prophylactic ICD implantation in individuals at high risk of MVP-related SCD. Some experts have suggested using electrophysiology studies for risk stratification in AMVP patients, and if sustained VT is present, ICD implantation is recommended for primary prevention [109].

Before the use of antiarrhythmic drugs, catheter ablation, and ICD, several small data studies have suggested that AMVP could be treated surgically with a lower rate of postoperative arrhythmias and antiarrhythmic drug discontinuation or reduction, even in the absence of significant MR [112,113].

Making treatment choices for AMVP must prioritize an individualized approach, considering the patient’s specific clinical characteristics and consultation with a healthcare professional experienced in managing MVP and arrhythmias. Further research is needed to establish more robust guidelines for the management of AMVP.

## 3. Conclusions

This review suggests that AMVP constitutes a noteworthy factor contributing to SCD. The risk factors associated with SCD in the context of AMVP encompass anomalies and detected abnormalities in electrocardiography (longer QTc interval and T-wave inversion, etc.), morphological patterns on TTE (like extensive myxomatous degeneration leading to substantial leaflet redundancy, and excessive leaflet length and thickness, MAD), as well as LGE on CMR, which offer potential avenues for the initial risk stratification among individuals diagnosed with MVP. Subsequent prospective investigations could be undertaken to validate and incorporate established parameters that assist in identifying MVP patients with an elevated susceptibility to encounter VAs and SCD, potentially proving advantageous for the implementation of primary prevention therapies.

## Figures and Tables

**Figure 1 diagnostics-13-02868-f001:**
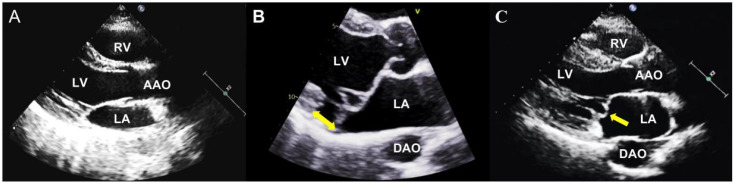
The mitral valve with and without MVP. (**A**) The parasternal long-axis view shows a normal mitral valve apparatus; (**B**) The parasternal long-axis view shows an MVP with MAD (yellow arrow); (**C**) The parasternal long-axis view shows an MVP without MAD (yellow arrow). LA: left atrium, LV: left ventricle, RA: right atrium, RV: right ventricle, AAO: ascending aorta, DAO: descending aorta, MAD: mitral annular disjunction.

**Figure 2 diagnostics-13-02868-f002:**
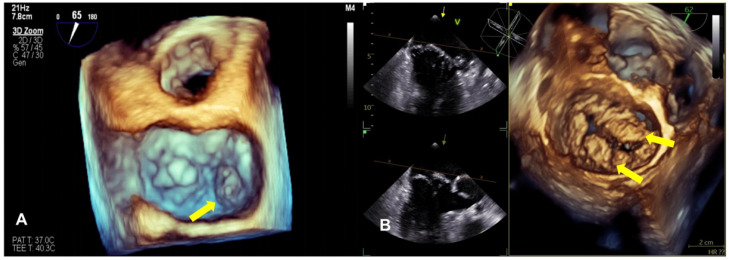
The panels show a 3D TEE surgical view of mitral valve prolapse. (**A**) This image shows the dislocation of the posterior mitral leaflet (P3) during systole (yellow arrow). (**B**) This image is typical of Barlow syndrome, which presents as bileaflet prolapse (yellow arrow).

**Figure 3 diagnostics-13-02868-f003:**
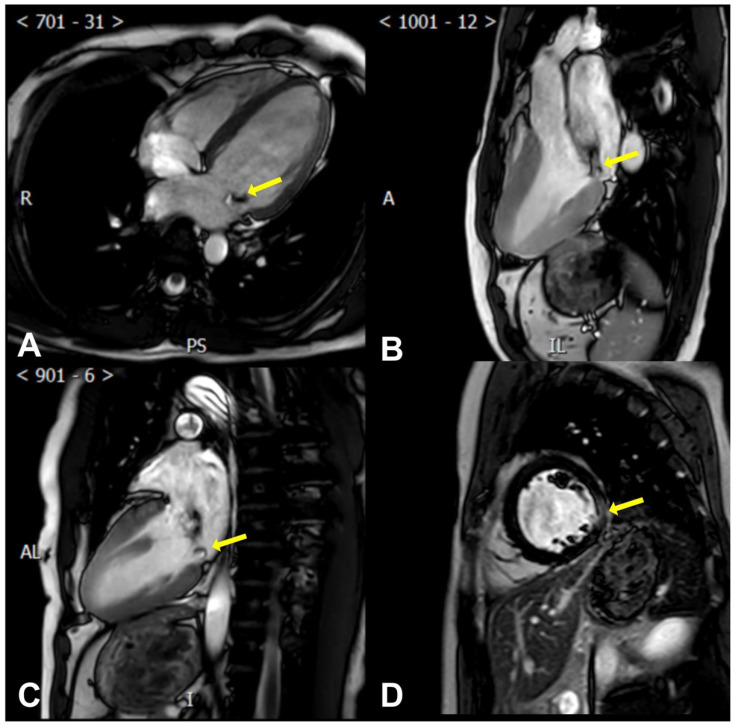
Cardiac magnetic resonance findings of AMVP. (**A**) The posterior mitral leaflet prolapse (yellow arrow) is seen in a perspective resembling the apical four-chamber view; (**B**) The posterior mitral leaflet prolapse (yellow arrow) is seen in a perspective resembling the apical three-chamber view; (**C**) The posterior mitral leaflet prolapse (yellow arrow) is seen in a perspective resembling the apical two-chamber view; (**D**) Late gadolinium enhancement of the left ventricular lateral wall (yellow arrow) is seen in a perspective resembling the parasternal short axis view.

**Figure 4 diagnostics-13-02868-f004:**
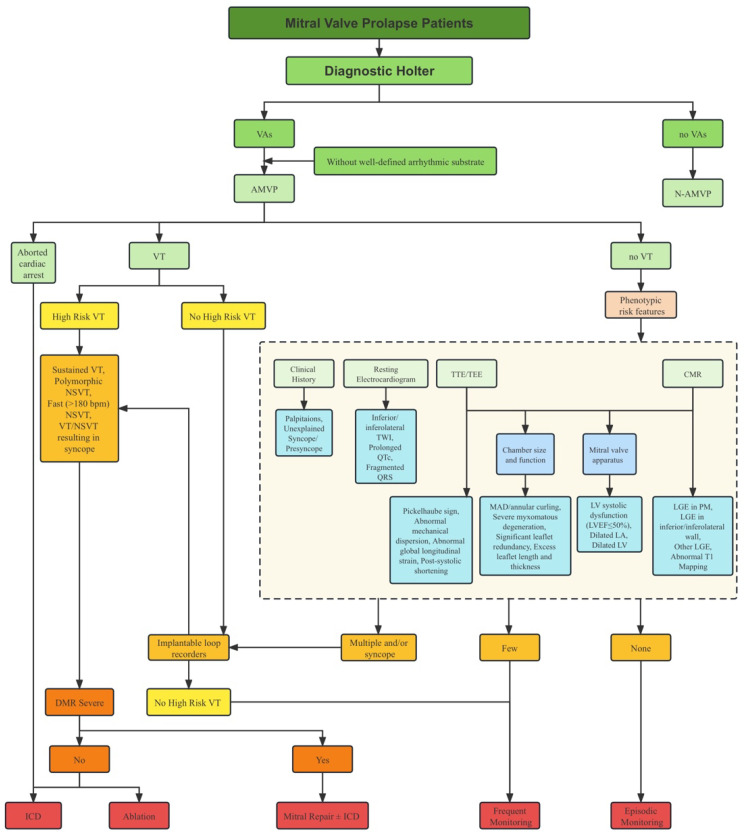
Risk stratification scheme. Approach for risk stratification for complex VA or SCD in AMVP patients. CMR, cardiac magnetic resonance; CT, computed tomography; DMR, degenerative mitral regurgitation; ICD, implantable cardioverter defibrillator; LA, left atrium; LGE, late gadolinium enhancement; LV, left ventricle; MAD, mitral annular disjunction; NSVT, non-sustained ventricular tachycardia; PM, papillary muscle; TTE, transthoracic echocardiography; TEE, transesophageal echocardiography; TWI, T wave inversion; VT, ventricular tachycardia.

**Table 1 diagnostics-13-02868-t001:** The 2022 EHRA classification of MVP-related VAs. MVP, mitral valve prolapse; VA, ventricular arrhythmia; PVC, premature ventricular contraction; NSVT, non-sustained ventricular tachycardia; VT, ventricular tachycardia.

Classification of MVP-Related VAs	
High risk	Sustained VT not originating from the right or left ventricular outflow tract
	Spontaneous polymorphic NSVT
	Rapid NSVT monomorphic (>180 bpm) has been associated with subsequent excess-mortality
Intermediate risk	Polymorphic PVCs
	NSVT monomorphic, of lower rate (<180 bpm)
	Highly frequent or complex PVCs (bigamy and couplets)
Low risk	Patients with frequent PVCs but not complex VAs (and no morphological higher risk features)

## Data Availability

Not applicable.

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
