# Peer review of "Arrhythmic Mitral Valve Prolapse: A Comprehensive Review"

_diagnostics, 2023, doi:10.3390/diagnostics13182868_

Round 1
Reviewer 1 Report
GENERAL COMMENTS
This Review covers an important topic “arrhythmic MV prolapse” that is generating an important debate. It is a comprehensive review including background, epidemiology, diagnosis with several modalities and clinical implications.
The Review is complete, interesting and well written, however some parts of the text can be improved.
SPECIFIC COMMENTS
Abbreviation
arrhythmic MV prolapse: A-MVP : check in introduction (previous abbreviation only in the abstract).
Epidemiology:
Pag 2 : lines 66-67. The prevalence of VAs and AMVP is not only partially defined because lack of ECG and Holter monitoring (as reported by Authors), but also because multimodality imaging determines different definition of MV patterns and populations of studies are heterogeneous.
Please quote: Deni Kukavica et al : Arrhythmic Mitral Valve Prolapse: Introducing an Era of Multimodality Imaging-Based Diagnosis and Risk Stratification. Diagnostics 2021.
Electrocardiography
Pag 4: line 144. And those with MR tend to show more PVCs …
This concept should be more detailed because arrhythmias may be more prognostically severe in cases with mild, end-systolic MR and bileaflet MVP. The severity of MR does not correlate with the arrhythmic burden.
MULTIMODALITY: echocardiography
Pag 5 and 6.
Insert brief comments inside this very complete chapter :
a) The importance of echo in defining the temporal occurrence of MVP and MR (typically end-systolic vs holosystolic..)
b) A part the Pickelhaube sign, the coexistence of “curling” of the LV postero-lateral wall has also a relevant diagnostic role.
CT and MRI versus ECHO and MAD
Faletra et al (REF 36 and 37) recently describe and discussed why MAD is still a difficult definition with imaging. “Although the posterior hinge line is somehow detached by the myocardium, it maintains its own ‘‘normal’’ position. In systole, as the posterior myocardium contracts, the hinge line slides backward, and MAD becomes visible. Which structure/tissue fills the gap between the hinge line and the crest of ventricular myocardium is unclear” In several cases there is a normal insertion of posterior leaflet in diastole, and its juxtaposition on the atrial wall in systole forming the pseudo MAD pattern (uncorrectly diagnosed as MAD).
Moreover the cut-off of the “normal” minimal MAD is different when evaluated by the different modalities and CT shows a 2mm MAD in the majority of normal heart. REFERENCE: Toh et al : Prevalence and extent of mitral annular disjunction in structurally normal hearts: comprehensive 3D analysis using cardiac computed tomography European Heart Journal - Cardiovascular Imaging (2021) 22, 614–622
Therefore due to sensitivity and specificity of the different modalities the cut-off of MAD with ECHO, TEE, CT and MRI has not been clearly defined (2 mm , 4 mm, 6 mm)
Maintain REF 112 and please quote also :
Mantegazza et al Mitral Annular Disjunction in a Large Cohort of Patients With Mitral Valve Prolapse and Significant Regurgitation. JACC Cardiovasc Imaging. 2019 Aug 8. pii: S1936-878X(19)30613-8. doi: 10.1016/j.jcmg.2019.06.021.
FIGURES
Fig 2 : this is not a typical case with bileaflet prolapse or Barlow. The Figure may be more clear if these images that represent a case of FED are compared (side by side) to a typical 3DTEE of a Barlow patient.
Fig 3 : Panel D is clear while Panels A, B and C are not of good quality.
Fig 4: upper line: ECG and clinical history must preceed Imaging techniques.
Please revise the entire diagnostic work up also on the basis of previous similar articles (or adapting this proposal to the flow chart proposed by the group of Padova, Basso and Iliceto).
REFERENCES
Insert the new references as reported in comments
Good.
Reviewer 2 Report
This is comprehensive review of arrhythmic mitral valve prolapse with more than 119 references. However, the references should be chosen more accurately.
There are some important issues to be clarified:
1. The title of the manuscript is “Multimpdal imaging of AMVP”. The Authors however, apart from imaging, cover also epidemiology, detailed description of postulated mechanism of ventricular arrhythmias, clinical manifestation and management of AMVP. I think the title is rather misleading.
2. I also have some doubts about references and their attribution. I would consider also a critical review of the way the references are used e.g
- Line 24: “While numerous studies have considered the vast majority of MVP to be benign, the prognosis for MVP varies widely and is an ongoing debate” – the references cited were published in 1987 and 1995 not reflecting the “ongoing debate”
3. Line 36 is on epidemiological data: metaanalisis by Han is correctly cited together with a case report - please clarify
4. Line 58-59 : “it has been suggested that A-MVP has a higher prevalence in females” and one of the references is an another case report – the pioneering one on Pickelhaube sign not on AMVP prevalence in females.
5. Line 61-62 a nationwide US autopsy study conducted between 2000 to 2018 found that gender distribution was equal” refer to ref on speckle-tracking echocardiography not on autopsy (22)
6. Line 64-67 on “The prevalence of VAs varies among MVPs…. “The references cited are from the seventies and the eighties last century.
7. In the chapter on pathology I would consider replacing the term posterior valve to posterior leaflet
8. Line 96 should be change to mitro-aortic instead of micro-aortic
9. Line 123-124 refer to paper by Han. In the original paper the 35% patients with MVP and SCD or cardiac arrest were given. Mot AMVP or SCD
10. In line 131-132 the authors describe ecg findings in AMVP and cite Ermakov’s paper on Left ventricular mechanical dispersion on echo
11. Same error in line 135-136 on negative T-waves, citing a pepar on echo by Carmo et al
12. There are date on fragmented QRS in MVP patients that could be referred to e.g. Kaya Ü, Eren H. Fragmented QRS may be associated with complex ventricular arrhythmias in mitral valve prolapse. Minerva Cardioangiol. 2020 Dec;68(6):577-585. Instead of the paper on fQRS in coronary artery disease.
13. Line 146-147 why ref 52 is cited. It has nothing to do
14. Line 242 – an error in the reference
15. Lines 263-265 please check the reference
16. Line 309: persistent VT? Please check
17. Line 322 – there are new European guideliens on SCD
18. Please check also ref to the chapter on MAD. While describing CMR as the gold standard the Authors cite a paper on echo by Manganaro et al Echocardiographic reference ranges for normal non-invasive myocardial work indices: results from the EACVI NORRE study[J]. European Heart Journal-Cardiovascular Imaging, 2019, 20(5): 582-590.
19. I wold question the reliance of the Authors on EPS in risk stratification of AMVP patient. , The role of EPS in identifying high-risk MVP is limited. And the statement that “EPS
should be considered if complex VAs are documented” is not true.
20. In the part on management the authors say that: “there is a lack of specific guidelines or consensus statements regarding the management of A-MVP”. However, there is a consensus released by EHRA in 2022. Strikingly, the Author cite it in different part of their manuscript as ref 17
21. Line 400 - new-generation antiarrhythmic drugs, what do the Authors mean?
Regarding Figures:
Figure 1 – please remove 4Ch view (part D) or add the information from the text (line 196 that the apical four-chamber view should not be used in diagnosis.
Figure 3 – please specify what frequent monitoring means. Should we monitor frequentlu the patient with MVP and MAD of 3 mm? . It is very uclear
1.
Minor remarks:
1. unnecessary capital letters in some references e.g 26., 29
2. typing errors in Figure 4 - should be prolonged QTs
The are some typing errors as mentioned above
Round 2
Reviewer 1 Report
No further suggestions
The new version markedly improved
Quality is ok
Author Response
Dear reviewer,
On behalf of my co-authors, I would like to express our heartfelt gratitude for your positive recognition of our manuscript.
Reviewer 2 Report
The Authors made appropriate corrections in the text. They also curated references by removing misquoted papers and adding the newest research, guidelines and consensus papers. The figures were also improved.
However, there are some points to be clarified:
1. Even though the authors claim to have corrected the text, in line 434 the referral to new-generation antiarrhythmic drug persisted.
2. the following sentence (line 391-394) seems unclear: " It is noteworthy that despite the propensity of MVP to cause life-threatening VAs, 391 the current guideline from the European Society of Cardiology concerning VAs and SCD 392 does not include the absence of explicit criteria for risk stratification and management 393 recommendations for VAs or SCD related to MVP[109].
3. There are also several annotations that "Reference source not found". e.g. line 93
4. Regarding the line 25 - I fully agree that there is an ongoing debate on prognosis in MVP. In the Round 1 of the review process I mentioned that by citing old references only the Authors do not prove the "ongoing" debate.
The English language is fine
